# The feasibility of a telephone-based weight loss intervention in rural Ohio: A pilot study

Xiaochen Zhang[1,2], Zachary L. Chaplow[3], Jessica Bowman[3], Abigail Shoben[4], Ashley S. Felix[2], Victoria R. DeScenza[3], Megan Kilar[3], Brian C. Focht[3], Electra D. Paskett[1] *

**1** Department of Internal Medicine, Division of Cancer Prevention and Control, The Ohio State University, Columbus, Ohio, United States of America, **2** Division of Epidemiology, College of Public Health, The Ohio State University, Columbus, Ohio, United States of America, **3** Department of Human Sciences, Kinesiology, The Ohio State University, Columbus, Ohio, United States of America, **4** Division of Biostatistics, College of Public Health, The Ohio State University, Columbus, Ohio, United States of America

owThese authors contributed equally to this work.
* electra.paskett@osumc.edu

**Data Availability Statement:** All relevant data are within the paper and its Supporting information files.

## Abstract

### Background

Weight loss through lifestyle modification can produce health benefits and may reduce cancer risk. The goal of this study was to examine the feasibility of and adherence to a 15-week telephone-based weight loss intervention in rural Ohio, an area with high rates of obesity.

### Methods

This pilot 2-arm randomized controlled study was designed for rural Ohio residents who were overweight or obese. Eligible participants were 2:1 randomly assigned to either a 15-week weight loss intervention group or active control group. The weight loss intervention group received weekly telephone sessions to improve healthy diet and increase physical activity. The active control group received education brochures with information on physical activity and dietary guidelines. Feasibility was defined as at least 80% of participants completing the follow-up surveys, and acceptable adherence was defined as the percentage of participants in the weight loss group who attend ≥75% of weekly telephone sessions.

### Results

A total of 423 individuals entered the online screening survey, 215 (50.8%) completed the survey, and 98 (45.6%) of those were eligible. Forty eligible individuals were enrolled and randomly assigned to the weight loss group (n = 27) or active control group (n = 13). The average age of the weight loss group was 49 (SD = 10) years, and 89% were female. The average age of the active control group was 51 (SD = 9) years, and 92% were female. Feasibility was demonstrated: 90% of participants completed the online follow-up surveys at 15-weeks. Among participants in the weight loss group, 22 out of 27 (81.5%) completed the 15-week intervention, the average number of sessions attended was 9.7 (64.9%). Adherence to the intervention was rated as acceptable among almost half of the group (48.1%).

**Funding:** This research was supported by an American Institute for Cancer Research grant. This work was supported by the National Cancer Institute (F99CA253745 to XZ). The funders had no role in study design, data collection and analysis, decision to publish, or preparation of the manuscript.

**Competing interests:** Electra Paskett would like to disclose that she has grant funding for work outside of this project from the Merck Foundation, Genentech and Pfizer. This does not alter our adherence to PLOS ONE policies on sharing data and materials. All other authors report there are no conflicts of interest."

## Conclusions

Feasibility of a 15-week telephone-based weight loss study among rural residents with over-weight/obesity were determined. A future study will test this intervention for weight loss efficacy.

## Introduction

Health disparities in rural populations are significant. About 97% of the land mass of the United States (US) is located in rural counties, and 60 million Americans (about 19%) live in these areas [1]. Residents in rural areas experience higher rates of obesity, obesity-related comorbidities and mortality, and higher cancer incidence and mortality than residents in urban areas [2,3]. Although socioeconomic and environmental disadvantages, including lower incomes, limited resources and availability related to health and health care, transportation barriers, geographic isolation, and lack of awareness, are linked to health disparities, obesity and poor health behaviors are the major modifiable contributors to health disparities in rural populations [4,5].

Obesity is the leading preventable cause of cancer [6]. About 40% of cancer cases and 14–20% of cancer-specific deaths are attributed to overweight and obesity [7]. Obesity leads to abnormal and excessive adiposity accumulation, which negatively impacts metabolic and inflammatory processes that promote cell growth, carcinogenesis, and tumor promotion [8,9]. Additionally, the dysfunctional adipose tissue promotes tumor progression and metastasis [8,10]. Due to these unfavorable physiological modifications, obesity is known to elevate the risk of 13 cancers [8,11,12].

Lifestyle weight management interventions have demonstrated metabolic benefits on insulin, lipid profiles, and improvements in body composition [13–15]. These outcomes are consistent with the hypotheses that weight loss, physical activity, and a healthy diet may potentially attenuate the negative effects of obesity and reduce cancer risk and premature mortality [16,17]. Evidence-based lifestyle weight loss interventions have been implemented in clinical settings; however, they have primarily focused on and utilized resources in urban areas [18–20]. Replicating intervention strategies from these programs may not be feasible or effective for rural populations [21,22]. Indeed, it is likely that extant weight management approaches would need to be significantly redesigned to address the needs of rural-dwelling communities, while also exploring alternative delivery modalities for increasing accessibility and scalability of interventions [23,24].

Moreover, rural residents reliably exhibit lower self-efficacy in managing healthy diet and exercise (i.e., one's belief in their capability to execute healthy eating/exercise behavior); especially when there are barriers and minimal social support from family and friends to promote a healthy lifestyle [21,22]. Self-efficacy and social support are critical theory-based determinants of successful weight loss efforts, given their role in supporting expectations, setting and achieving goals, increasing daily physical activity, establishing healthy dietary habits, and promoting persistence in managing barriers [25,26]. However, theory-driven behavioral interventions integrating tailored strategies to improve self-efficacy for lifestyle change and social support for rural populations are understudied [23,24,27]. Furthermore, critical feasibility and implementation variables, such as adherence and behavior change techniques have been infrequently reported. Therefore, this study examined the feasibility of a telephone-based weight loss intervention integrating tailored strategies to improve self-efficacy and increase social

support among rural Ohio residents to improve healthy diet and physical activity and maintain a healthy weight.

## Materials and methods

### Study design

This was a single-blind, 2-arm randomized controlled pilot study designed to determine the feasibility of a 15-week telephone-based weight loss in rural populations with overweight/obesity. Briefly, feasibility in this study was defined as ≥80% of the participants completing the online follow-up surveys, and acceptable adherence was defined as the percentage of participants in the weight loss intervention group engaged in ≥75% of the weekly sessions. This study was approved by The Ohio State University Institutional Review Board (2021C0033) and registered on ClinicalTrial.gov (NCT05040152).

### Study population and recruitment

Participants were recruited through Facebook advertising, flyers, a list of participants from a previous study, family/friends referral, and ResearchMatch or StudySearch. Individuals who were interested in the study were consented and then completed eligibility screening by phone or online survey using the Research Electronic Data Capture (REDCap) web application. Participants meeting the following criteria were eligible for the study: 1) lived in a rural Ohio county (according to the Rural-Urban Continuum Codes); 2) BMI $\geq 25kg/m^2$; 3) 20–64 years old; 4) not currently participating in any weight loss intervention; 5) not meeting the 2018 Physical Activity Guidelines for Americans (150 min/week of moderate-intensity exercise or 75 min/week of vigorous exercise); 6) able to walk two city blocks (about 400 steps); and 7) able to speak and read English. Individuals who: 1) had a prior cancer diagnosis (except non-melanoma skin cancer) or severe medical conditions such as unstable cardiovascular disease or digestive disorders that would preclude physical activity and dietary interventions; 2) were pregnant or nursing; and 3) were unable to give informed consent were excluded from the study.

Eligible participants were grouped by their county of residency locations and scheduled an initial in-person visit at either Pike County YMCA (Waverly, OH), Jackson County Health Department (Jackson, OH), or Hardin County YMCA (Kenton, OH). For individuals who self-reported to have pre-existing health conditions (e.g., heart problems, insulin-dependent diabetes, poorly controlled blood pressure, other severe diseases), medical clearance from their respective physicians was obtained before the initial in-person visit.

### Randomization

After scheduling the initial in-person visit, a total of 40 participants were enrolled and randomized in a 2:1 fashion to either the weight loss intervention group (Telephone-based health counseling, n = 27) or the active control group (Health education, n = 13). All participants were asked to provide written informed consent during the initial in-person visit before any study activities began.

### Telephone-based health counseling group

The 15-week telephone-based, multi-component approach to weight loss intervention was adapted from the Look AHEAD study and Healthy Living and Eating Program [18,28]. We modified the in-person sessions to be delivered as a remotely accessible, telephone-based intervention, thus reducing the burden of travel and tailored the program to the rural population.

S1 Table includes intervention components and supplementary resources provided to the weight loss intervention group. Participants from the weight loss intervention group received weekly telephone-based health counseling sessions (30–45 minute) led by health coaches with a background in exercise physiology and behavioral weight loss interventions for 15 weeks with the goal of each participant to achieve an approximately 7% weight loss from baseline. The dietary, physical activity, and behavioral support domains were all addressed during each of the counseling sessions. The weekly telephone sessions followed standardized lesson plans, while critically incorporating the necessary flexibility for tailoring and responding to more immediate participant needs. During each session, instructions on and recommendations for independent dietary and physical activity were made, along with instruction on the use of complementary self-regulation strategies.

The Behavior Change Technique Taxonomy version 1 (BCTTv1) was created to address the need for standardizing behavior change intervention methods and was used to code behavior change techniques (BCTs) [29]. Recent advances in linking BCTs with theories of behavior change have allowed for the investigation of possible mechanisms of action (MoAs) [30]. Therefore, to facilitate replicability and enhance reporting precision of both study arms, S2 Table provides the unique applications, BCTs and proposed MoAs targeted by the weekly telephone health counseling. No BCTs were coded for resources that were provided without additional advice, instruction, or demonstration from health coaches.

The targeted behavioral modifications for weight loss included 1) a modest restriction in caloric intake (500–1000 kcal/day) that gradually progressed towards a personalized, target daily caloric intake goal (1200–1800 kcal/day); and 2) a concomitant increase in energy expenditure via physical activity (including aerobic exercise and resistance training) (Table 1). To tailor the intervention to address the specific needs of rural participants, the health coach focused on building healthy behaviors using local resources available to each participant, suggesting behavioral techniques to overcome individual barriers, and encouraging the establishment of a support system with family and friends. Participants in the telephone-based weight loss intervention group received a Fitbit and a weight scale to track their physical activity and body weight, respectively, and were asked to self-monitor dietary intake through MyFitnessPal during the 15-week intervention period. They also received a lifestyle modification manual to guide behavioral changes with weekly goals.

**Table 1. Characteristics of individuals who completed the online screening survey, overall and those who were consented to the study vs. those were screened but excluded from the study.**

| Characteristics | Overall | Consented | Screened but Excluded | P |
|---|---|---|---|---|
| | n = 215 | n = 41 | n = 174 | |
| Age | 49.56±9.65 | 49.24±9.23 | 49.63±9.76 | 0.82 |
| BMI* | 37.92±9.74 | 35.4±5.85 | 38.5±10.37 | 0.07 |
| Gender | | | | 0.9 |
| Male | 25 (11.6%) | 5 (12.2%) | 20 (11.5%) | |
| Female | 190 (88.4%) | 36 (87.8%) | 154 (88.5%) | |
| Health conditions | | | | |
| Heart diseases | 8 (3.7%) | 1 (2.44%) | 7 (4.02%) | 0.63 |
| Insulin-dependent diabetes | 9 (4.2%) | 1 (2.44%) | 8 (4.6%) | 0.54 |
| Poorly controlled hypertension | 22 (10.2%) | 5 (12.2%) | 17 (9.8%) | 0.65 |
| Other severe disease | 10 (4.7%) | 0 (0%) | 10 (5.8%) | 0.12 |

*BMI calculated according to self-reported weight and height.

**Dietary component.** During the weekly telephone sessions, participants received dietary recommendations tailored to their current weight and weight loss target. The specific dietary objectives were consistent with recommendations from the American Institute of Cancer Research (AICR) [31]. The AICR nutrition guidelines were selected to be implemented in the dietary portion of the intervention because they are the most comprehensive, scientifically based cancer prevention guidelines available, and the education materials available from AICR in support of their guidelines are clear, easy-to-follow, and very consumer-friendly.

The dietary intervention encouraged reductions in portion size and caloric and fat consumption together with a gradual transition from an animal-based diet to a more plant-rich diet while still incorporating animal foods, including milk and meat, with an emphasis on monitoring food proportion and portion size. Specifically, the dietary component included: 1) reduction in energy intake by 500–1000 kcal per day; 2) reduction in total fats to 25–30%, saturated fats to 7%, and protein to 15% of total calories; 3) increase in fruit and vegetable consumption to 5 servings per day; and 4) intake of 3 or more servings per day of whole grains and a gradual increase to at least 25 grams of dietary fiber per day.

The weekly sessions on dietary intake used a motivational interviewing approach [32]. This technique has been extensively used in substance abuse and lifestyle behavioral counseling and has been applied to exercise and diet-related behavior change interventions [33–36]. Motivational interviewing has been demonstrated to be an effective method to promote behavior change and enhance weight management in various populations [37,38]. Additionally, participants received tailored dietary strategies built upon many of the cognitive-behavioral self-management techniques utilized in the behavioral exercise counseling component, including self-monitoring, building self-efficacy, goal-setting, and anticipating and managing barriers to dietary behavior change. Dietary consultations were provided by a registered dietitian to participants as needed, in addition to the weekly sessions.

**Physical activity component.** An evidence-based physical activity plan, consisting of a combination of home-based aerobic and resistance exercises was described through a lifestyle modification manual and promoted during the telephone counseling sessions. The study goals for physical activity were to achieve 150–200 min/week of moderate-intensity aerobic physical activity and 2–3 sessions/week of resistance training, according to physical activity guidelines [16]. The aerobic physical activity recommendation consisted of gradually progressing from 10–30 minutes of exercise on most, if not all days of the week, with a primary focus placed upon brisk walking as the primary accessible mode of aerobic exercise for each participant. However, participants were encouraged to engage in their preferred choice of exercise mode that they were able and willing to perform, including brisk walking, running, or cycling. The home-based progressive resistance exercise involved performing 1–3 sets of 8RM-12RM repetitions of various exercises (e.g., sit-to-stand, standing leg curl, seated leg extension, chest press, lateral raise, bent-over row, straight-arm pulldown, arm curl, triceps pushdown, and chair abdominal curls) using body weight and resistance bands. All exercise prescriptions were appropriately modified for safe and effective home-based exercise. For example, for participants who were not familiar with aerobic or resistance exercises, the health coach prescribed a relatively low dose and intensity of exercise in the first few weeks, with the intention to allow participants to gain confidence in exercising, promote gradual integrating of exercise into their schedule, and encourage them to set practical and achievable intermediate exercise goals that would not be overwhelming. For participants who had certain health conditions (e.g., knee osteoarthritis, back pain), the health coach would modify the resistance exercises to accommodate participants' needs.

During the telephone counseling in the first two weeks, health coaches went through the exercises described in the lifestyle modification manual with each participant, and if needed,

modifications of the exercises were provided. Online *YouTube* videos were available to participants demonstrating safe and effective techniques for each of the prescribed exercises, providing examples for alternative exercise options, and providing tips for proper exercise form. These videos have been successfully implemented to facilitate home-based exercise in prior lifestyle interventions [39]. In order to maintain a safe and gradual progression in exercise stimulus, each week, participants were encouraged to increase their exercise frequency, duration, and/or intensity until successfully attaining the target physical activity volume goal.

**Behavioral component.** The behavioral support portion of each telephone counseling session centered on promoting self-management strategies tailored to rural populations using theory-informed and contemporary principles of the cultural-tailoring approach [25], focused on motivation, relapse prevention, emotional distress, time management, and overcoming barriers. Each session included a review of the previous week's activity, planned weekly lessons, dietary and exercise goals for the subsequent week, and strategies tailored to each participant to improve self-efficacy, social support, and barrier management. Additionally, group-mediated activity through group messages, following the weekly telephone counseling, was integrated with the behavioral component to promote independent adherence to lifestyle modification. The group messaging was supervised by the health coaches and primarily used as a means of communication between participants. This ancillary supportive approach was used to facilitate group-mediated barrier problem solving and as a means for participants to share successes and challenges and provide encouraging feedback to their peers.

The objective of the behavioral component was to promote motivation, self-efficacy and social support for adopting lifestyle modifications, while gradually facilitating participants' transition towards successful independent self-regulation of a healthy diet and physical activity. The behavioral component focused on the acquisition and practice of self-regulatory skills, in conjunction with a continuous problem-solving model of behavior change, to empower participants to exert greater control over their behavior, cognition, and environment. The behavioral component was designed to a) increase health knowledge of the benefits of physical activity and dietary modifications; b) enhance self-efficacy and positive outcome expectancies through the promotion of a series of successful experiences in changing exercise and eating behavior; and c) improve self-regulation of exercise and eating behaviors through the use of goal-setting, self-monitoring, stimulus control, cognitive restructuring, and barrier problem-solving strategies. The efficacy of this approach for promoting the adoption and maintenance to exercise and dietary behavior change has been demonstrated in prior randomized controlled trials [39,40].

## Health education control group

Information and supplementary resources provided to the active control group are included in S1 Table. At the initial in-person visit, participants who were randomized to the active control group received education brochures describing the AICR physical activity and dietary guidelines. Participants were asked to meet the 2018 Physical Activity Guidelines for Americans (150 min/week of moderate-intensity exercise or 75 min/week of vigorous exercise), increase fruit and vegetable intake, reduce fat intake, reduce sugary drinks, increase consumption of whole grains, and increase water intake. An exercise manual with access to online *YouTube* videos was available with examples of proper exercise forms for strength training. Participants were provided with self-monitoring resources for body weight, dietary intake, and physical activity through MyFitnessPal or a paper log. To increase retention and provide motivation and strategies for behavioral change in the active control group, participants received a Fitbit and the lifestyle modification manual at the end of the study.

## Measurement

All participants were asked to complete two in-person assessments: baseline at the initial visit and the 15-week at the end-of-study visit. The assessments described below were conducted by research staff for all participants at both visits. All data were collected and stored using the REDCap secure web-based application hosted at OSU.

**Anthropometric assessment and body composition.** Body weight (kg) and height (cm) were assessed by research staff using a digital scale and scale-mounted stadiometer to calculate BMI (kg/m$^2$). Body composition, including fat mass (kg), percent fat, lean mass (kg), and percent lean mass were measured using a three-dimension (3D) body scanner, Styku S100 (Styku L.L.C., Los Angeles, CA). The Styku S100 scanner uses an infrared camera built into a tower and a rotating platform to capture body circumference and volumes. Participants stood on the platform with their arms positioned in the A-pose, and then the platform rotated clockwise. The Styku S100 scanner uses Microsoft Kinect Fusion software to recognize body landmarks and calculate and visualize various body measurements [41]. The Styku S100 scanner is portable to use in rural community-setting and produces reliable measures of body composition compared with dual-energy X-ray absorptiometry [41].

**Lipid profile.** Total cholesterol (TC), triglycerides (TG), high-density lipoprotein cholesterol (HDL-C), and low-density lipoprotein cholesterol (LDL-C) were obtained by fasting capillary blood sampling from fingerstick and analyzed using Cholestech LDX (Hayward, CA). The Cholestech LDX has demonstrated accuracy and precision in assessing blood lipids [42,43]. It is a point-of-care assessment to screen abnormal blood lipids in clinical, community, and other health promotion settings [44,45]. The Cholestech LDX provides lipid profiles result within 5 minutes, which allows research staff and participants to receive results during the same visits.

**Inflammatory markers.** Participants also underwent fasting ($\geq$8 hr) blood draw for inflammatory analyses. Serum samples were stored at -80°C until assayed. C-reactive protein (CRP) concentration (mg/L), Interleukin (IL)-6 concentration (pg/mL), and Tumor Necrosis Factor (TNF)-$\alpha$ were quantified using Meso Scale Discovery (MSD) Platform (Meso Scale Diagnostics, Rockville, Maryland).

**Physical activity.** The objectively measured physical activity was recorded using the LIFE-CORDER Plus Accelerometer (Suzuken-Kenz, Inc. Nagoya, Japan) for seven days in the 1st and 15th week of the intervention [46]. Participants were provided verbal and written instructions on how to wear the ActivPAL and wore the monitor 24 hours per day for the seven consecutive days following the completion of each assessment visit. Monitors were then returned to trial staff via US postal service. Participants also completed self-reported Leisure-Time Exercise Questionnaires [47].

**Dietary intake.** Participants were asked to complete a 7-day food checklist following study assessments. This food checklist was developed by the National Cancer Institutes and comprised of a list of 32 foods [48]. This daily food checklist yields estimates of macronutrient intake, as well as intake by specific food groups. Participants in the weight loss intervention were encouraged to monitor dietary intake using MyFitnessPal (or paper log) during the trial. The food log and weekly summary of caloric and macronutrient intake were regularly monitored by their health coach and registered dietitian to track adherence.

**Self-efficacy and social support.** Participants completed self-reported Healthy Weight and Eating Self-Efficacy, Multi-dimensional Self-Efficacy scales [49–51], as well as adapted Social Support Inventory for Physical Activity (SSPA) and Eating Habits (SSEH) questionnaires [52,53].

**Physical fitness.** Participants also completed valid and reliable timed performance-related mobility tasks, including 400-meter walk and lift and carry tasks [54,55]. These assessments at baseline served as a reference to tailor exercise counseling for each participant.

## Statistical analysis

The feasibility of the study was defined as having at least 80% of participants complete the follow-up online surveys at week 15. The feasibility of the weight loss intervention was defined as the percentage of participants in the weight loss group who completed the 15-week intervention. With 27 participants in the weight loss group, if the true completion rate was 80%, we had 92.5% power to rule out the unacceptable 50% completion rate with 95% confidence (one-sided).

The acceptable adherence of the weight loss intervention was defined as the percentage of participants in the weight loss intervention group who engaged in at least 75% of the weekly telephone counseling sessions. Other feasibility assessments, such as the reasons for ineligibility, reasons for refusal among eligible participants, barriers to adherence to weekly sessions, and reasons for dropout, were also assessed.

Descriptive statistics, including age, gender, self-reported BMI, and health conditions, were reported overall and compared between those who consented to the study and those who met the eligibility criteria but were excluded. Among those who were enrolled in the study, demographic characteristics, baseline weight, body composition, and objective-measured physical activity were reported with means/standard deviations (SD) for continuous variables and frequencies for categorical variables by study group. The feasibility of the study and the weight loss intervention, and the adherence of the weight loss intervention (as defined above) were calculated.

## Results

Between September 23 to December 14, 2021, a total of 423 individuals entered and consented to complete the REDCap eligibility screening survey. All individuals were recruited from Facebook (n = 384), flyers (n = 9), postcards to participants from a previous study (n = 23), family/friends (n = 3), and ResearchMatch or StudySearch (n = 4). Among them, 215 (50.8%) individuals completed the eligibility survey, 111 were ineligible, five refused to participate, and one individual provided an incorrect contact number (Fig 1). Among the 98 (23.2%) individuals who met eligibility criteria, 24 individuals were unable to be contacted (average contact attempt = 5), three were not interested at this time, and 11 were not contacted as the study accrual was met. Among those who were eligible, 60 (61.2%) individuals were scheduled for in-person consent and baseline assessment. However, we had 19 (31.7%) no show at the scheduled baseline visit (n = 5 sick/medical reason; n = 3 COVID or COVID-related; n = 1 family emergency; n = 2 vehicle issues; n = 2 no time; n = 6 unable to contact). Among the 41 participants who consented, one was not eligible after weight assessment and was excluded from the study. A total of 40 participants enrolled in the study, 27 (67.5%) were randomized to the weight loss intervention, and 13 (32.5%) were randomized to the active control group.

There were no statistical differences in age, gender, BMI, and existing health conditions observed between those who consented to the screening survey and ultimately enrolled in the study vs. those who were screened but excluded from the study (Table 1).

Table 2 describes the baseline characteristics among participants in the weight loss intervention (n = 27) and active control groups (n = 13). The average age was 49 (SD = 10) years for the weight loss group and 51 (SD = 9) years for the control group. In both groups, most participants were female. At baseline, among participants in the weight loss group, the average

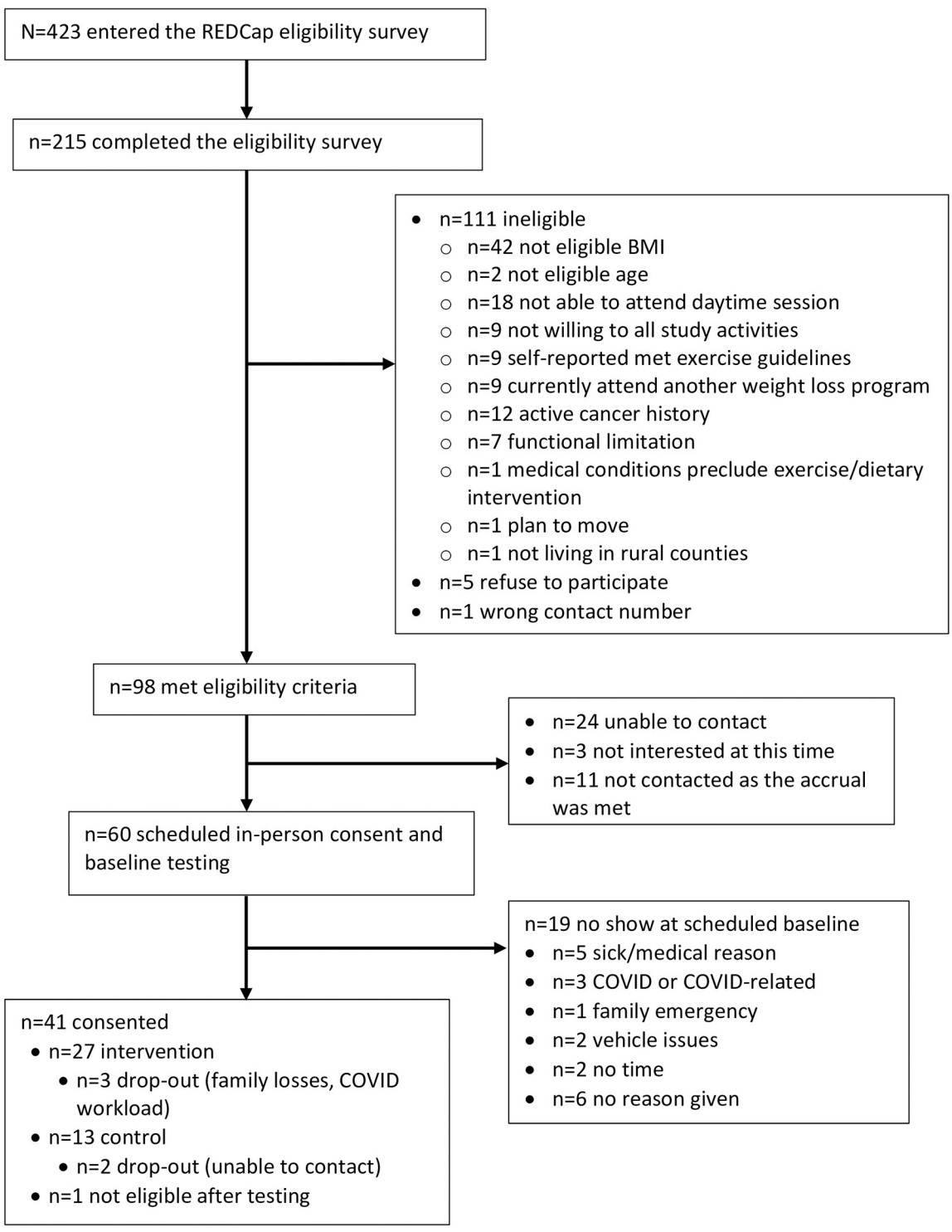

**Fig 1. HERO study flowchart.**

body weight was 228 (SD = 41) lbs., the average BMI of 37 (SD = 6) kg/m$^2$, and the average percent fat was 42 (SD = 5) %. Among participants in the control group, the average body weight was 229 (SD = 59) lbs., the average BMI was 39 (SD = 13) kg/m$^2$, and the average percent body fat was 41 (SD = 4) %.

**Table 2. Baseline characteristics of participants by study group.**

| | Telephone-based Health Counseling (n = 27) | Health Education Control (n = 13) |
|---|---|---|
| *Demographic* | | |
| Age, years | 49±10 | 51±9 |
| Female, n (%) | 24 (89%) | 12 (92%) |
| Male, n (%) | 3 (11%) | 1 (8%) |
| *Community center, n (%)* | | |
| Pike County Y.M.C.A. | 5 (19%) | 4 (31%) |
| Jackson County Health Dept | 11 (41%) | 3 (23%) |
| Hardin County YMCA | 11 (41%) | 6 (46%) |
| *Baseline Anthropometric* | | |
| Weight, lbs | 227±41 | 228±59 |
| BMI, kg/m2 | 37±6 | 39±13 |
| Fat mass, lbs | 96±20 | 88±19 |
| % Fat | 42±5 | 41±4 |
| Lean mass, lbs | 126±27 | 117±23 |
| % Lean | 55±5 | 56±4 |
| *Baseline Physical Function* | | |
| 400 Meter Walk Time, Seconds | 301±44 | 296±47 |
| Lift & Carry Time, Seconds | 9±2 | 10±2 |

In terms of feasibility, 35 out of 40 participants (88%, 95% CI = 0.78–0.98) completed the follow-up online surveys at Week 15. Among participants in the weight loss group, 22 out of 27 (82%) completed the 15-week intervention, and the average number of sessions attended was 10 (67%). Specifically, 5 participants attended the full 15 sessions, 8 attended 12–14 sessions, 5 attended 8–11 sessions, and 9 attended less than 8 sessions. Among the 14 participants who attended less than 12 weekly sessions, the common reasons for missed sessions included increased caregiver responsibility due to COVID, being too busy, work conflict (due to COVID), and illness. Adherence to the weight loss intervention was 48% (13 out of 27 participants attended at least 12 sessions). Among participants who dropped out of the weight loss intervention (n = 5, 19%), three were due to a COVID-19-related workload increase, one was due to a family loss, and one preferred a non-telephone-based approach. Among the 13 participants in the active control group, 2 (15%) were lost to follow-up.

## Discussion

Our results indicate that it was feasible to conduct a 15-week randomized controlled weight loss study among rural residents who were overweight/obese. We also found that it was feasible for rural residents to participate in a 15-week telephone-based weight loss intervention. Although efforts were taken to retain participants (e.g., in-person assessment at a community center close to participants, gift card incentives, and offering the active control group the same lifestyle manual and Fitbit as the weight loss group at the end of the study), we observed some dropouts in both groups. However, the dropout rate of our study was less than 20%, indicating that this study design is feasible for this population [56].

Rural residents, who experience disparities in obesity and cancer incidence and mortality, are a vulnerable population that can benefit from weight loss interventions. Most weight loss programs have focused on health-promoting resources available in urban areas [14,18–20].

Our study integrated a multi-component, evidence-based lifestyle behavioral change intervention, delivered by experienced health coaches, through telephone-based cognitive-behavioral counseling. By tailoring the intervention strategies to the rural-dwelling population, we were able to provide a remotely accessible telephone-based weight loss program for rural residents to address their unique needs [18,24,27,28]. Among recently completed weight loss interventions that targeted rural populations, few reported feasibility (83%) and acceptability (77–90%) [57,58]. Furthermore, their study populations and intervention delivery approaches were different from our study. One study targeted older adults (>65 years old) and utilized video conferencing for weight loss counseling [58]. Another pilot study recruited participants from a Weight and Wellness Center [57]. Participants from those studies may have a more flexible schedule that allows them to attend assigned sessions. They may also prioritize losing weight more than our population. In our study, participants had an average age of 50 years, with most still retaining daytime jobs. Although efforts were made to extend the flexibility of weekly telephone sessions and improve the communication between the health coach and each participant, the adherence of the telephone-based intervention was only 48.1%.

The disruption in daily life during the pandemic may have contributed to the low adherence to the weight loss intervention. Since the pandemic started, loss of daily routine, lack of access to groceries, stockpiling with highly processed foods, irregular eating due to stress, and financial burden related to unemployment from COVID-19 had negative impacts on health behaviors [59,60]. Increased anxiety and depression due to COVID, decreased physical activity, reduced fruit and vegetable intake, increased alcohol consumption, and increased tobacco intake have all been reported since the pandemic started [59,61]. During the study period, we noticed these negative impacts on each participant, and we understood that the priorities for our participants might shift to other aspects. Our health coaches provided strategies, including practicing mindfulness and offering resources for relaxation and guided meditation, to overcome COVID-related challenges. Future interventions should consider incorporating structured mindfulness practices to improve resilience and mental health for this population.

Due to the impact of the COVID-19 pandemic, we encountered some challenges in recruitment and retention, in addition to delays in receiving equipment and supplies. As we were unable to attend community events, our recruitment solely depended on Facebook advertising, flyers, and mailed postcards to participants from a previous study. Among individuals who met the eligibility criteria and were interested in the study, there was a two-week waiting period before the scheduled baseline in-person sessions. The two-week period allowed us to obtain medical clearance for those with medical conditions, however, it resulted in a high no-show rate of 31.7%. Common reasons for no-shows included sick/medical-issue, COVID or COVID-related issues, or family emergencies, all of which were reasonable due to the unprecedented pandemic disruption. Future studies may consider kick-off sessions to engage and motivate individuals, improve connections with potential participants and the community centers, and reduce the no-show rate.

This study has several limitations. It was a relatively small, homogenous sample of 40 participants who were all White and mostly women, in rural Ohio. Our findings may not generalize to more diverse populations in different geographic areas. Additionally, we found individuals who enrolled in our study had a lower BMI compared to individuals who were interested but not enrolled. Therefore, our study sample may not represent the general populations in rural areas. Moreover, some participants experienced cellphone reception issues that impeded them from receiving phone calls or group messages. With the new federal infrastructure investment in the next few years, especially the improvement of internet services, future studies may consider utilizing video-conferencing or other telehealth approaches to deliver remotely accessible weight loss programs for this population.

Our study has several strengths. First, to tailor the weight loss intervention to rural populations, the weekly sessions specifically focused on optimizing accessible resources, improving self-efficacy, and promoting social support to help rural populations overcome barriers, with the goal of increasing physical activity, improving healthy diet, and maintaining a healthy weight. Most of the participants lived in rural areas inundated by quick-service restaurants. Consequently, health coaches routinely provided strategic guidance to help participants successfully manage their dietary behavior and environment. For example, one participant reported that she had to frequently resort to fast-food restaurants, simply because there was no other option. As a substitution for the typical high-calorically dense meal choices, the coach suggested ordering a side salad and a 4-piece chicken nugget (about 500 kcal total) when necessary and combining this with the mindful eating and planning skills developed through the counseling sessions.

The importance of identifying the 'active ingredients' within behavior change interventions, is critical to advancing their dissemination and implementation. BCTs are the "observable, replicable, and irreducible components of interventions designed to alter or redirect causal processes that regulate behavior" [29]. In the weight loss intervention group, approximately, 33 of the 93 BCTs were used in the multi-component weekly sessions to promote adherence to dietary and physical activity recommendations and encourage active self-regulation (S2 Table). These sessions were delivered by trained health coaches experienced in lifestyle behavior change processes who possessed the capacity to adapt the intervention to participant need. BCTs provided as part of the weekly telephone counseling sessions represented 14 of the 16 BCT clusters from the BCTTv1 [29].

Additionally, our study included an in-house registered dietitian available to provide valuable guidance for the weight loss intervention group upon request on an as-needed basis. We noticed that many participants were not familiar with macronutrients. Besides individual nutrition counseling, the dietitian offered supplementary educational and visual aids to illustrate the macronutrient content of commonly consumed foods and to facilitate improvement in dietary decision-making. This was crucial for the rural population to improve their awareness, self-efficacy, and planning for healthy diet. We also provided the active control group with AICR physical activity and dietary guidelines, as well as the exercise manual and online exercise videos (S1 Table). Some of the individuals from the active control group were inspired by these materials and made behavioral changes, which showed the potential benefits of the dissemination of behavioral guidelines to rural populations.

Lastly, we were, to date, the first study to use a novel, portable 3D body scanner, Styku S100 (Styku LLC, Los Angeles, CA), to assess body composition, including fat mass and fat percentage, in rural community settings. Measures of body composition, especially body fat, are relevant to assessing cancer risk, as adiposity deposition is associated with inflammation and carcinogenesis [62,63]. Among the handful of completed or recently awarded lifestyle weight loss trials in rural populations, none measured body composition [64,65]. This might be due to the access burden of standard body composition measurement tools (e.g., DXA) in rural community settings. We also used a finger stick to measure lipid profile, which provided the results to each participant on site. This might be a potential point-of-care approach to assess disease risk factors in rural populations for future studies.

In conclusion, our pilot study provided important data and valuable experiences in conducting a telephone-based weight loss intervention among rural populations with overweight/obesity. In our study, a 15-week study to promote physical activity, healthy eating, we demonstrated that conducting a study to promote a healthy diet is feasible among rural Ohio residents. We also demonstrated that it was feasible for rural adults with overweight/obesity to participate in a 15-week telephone-based weight loss intervention. Further studies should

refine strategies to improve the uptake of the weight loss intervention, as well as evaluate the effect size of the telephone-based intervention to serve as preliminary data for a large-scale, randomized controlled trial to quantify the therapeutic effect of weight loss interventions and cancer risk in rural populations.

## Supporting information

**S1 Table. Study materials and supplementary resources by study group.**
(DOCX)

**S2 Table. Telephone-based health counseling intervention components, Behavior Change Techniques (BCT), and Mechanisms of Action (MoA).**
(DOCX)

**S1 File. Minimal data.**
(XLS)

## Acknowledgments

We acknowledge Dr. Justin C. Brown and Dr. Jennifer A. Ligibel for sharing intervention contents from the Healthy Living program. We appreciate supports from the rural community centers (Pike county YMCA, Jackson county Health Department, and Hardin county YMCA). This work could not have been completed without assistance from students from the Exercise Behavioral Medicine Lab at OSU.

## Author Contributions

**Conceptualization:** Xiaochen Zhang, Zachary L. Chaplow, Abigail Shoben, Victoria R. DeScenza, Brian C. Focht, Electra D. Paskett.

**Data curation:** Xiaochen Zhang, Zachary L. Chaplow, Jessica Bowman, Victoria R. DeScenza, Megan Kilar.

**Formal analysis:** Xiaochen Zhang.

**Funding acquisition:** Xiaochen Zhang, Victoria R. DeScenza, Brian C. Focht, Electra D. Paskett.

**Investigation:** Xiaochen Zhang, Zachary L. Chaplow, Jessica Bowman, Ashley S. Felix, Victoria R. DeScenza, Megan Kilar, Brian C. Focht, Electra D. Paskett.

**Methodology:** Xiaochen Zhang, Zachary L. Chaplow, Jessica Bowman, Abigail Shoben, Ashley S. Felix, Victoria R. DeScenza, Megan Kilar, Brian C. Focht, Electra D. Paskett.

**Project administration:** Xiaochen Zhang, Zachary L. Chaplow, Jessica Bowman, Victoria R. DeScenza, Megan Kilar.

**Resources:** Zachary L. Chaplow, Jessica Bowman, Victoria R. DeScenza, Megan Kilar, Brian C. Focht, Electra D. Paskett.

**Supervision:** Xiaochen Zhang, Abigail Shoben, Ashley S. Felix, Brian C. Focht, Electra D. Paskett.

**Writing – original draft:** Xiaochen Zhang, Electra D. Paskett.

**Writing – review & editing:** Xiaochen Zhang, Zachary L. Chaplow, Jessica Bowman, Abigail Shoben, Ashley S. Felix, Victoria R. DeScenza, Megan Kilar, Brian C. Focht, Electra D. Paskett.

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
