## [Decision Letter · Decision Letter 0]

4 Nov 2022

PONE-D-22-26251The Feasibility and Acceptability of a Telephone-based Weight Loss Intervention in Rural Ohio: A Pilot StudyPLOS ONE

Dear Dr. Zhang,

Thank you for submitting your manuscript to PLOS ONE. After careful consideration, we feel that it has merit but does not fully meet PLOS ONE’s publication criteria as it currently stands. Therefore, we invite you to submit a revised version of the manuscript that addresses the points raised during the review process.

We look forward to receiving your revised manuscript.

Kind regards,

Jamie Matu, Ph.D.

Academic Editor

PLOS ONE

Journal Requirements:

"This research was supported by an American Institute for Cancer Research grant to BF (https://www.aicr.org/).

This work was supported by the National Cancer Institute (F99CA253745 to XZ; https://www.cancer.gov/)."

Please state what role the funders took in the study.  If the funders had no role, please state: ""The funders had no role in study design, data collection and analysis, decision to publish, or preparation of the manuscript."" If this statement is not correct you must amend it as needed. 

"I have read the journal's policy and the authors of this manuscript have the following competing interest:

Electra Paskett would like to disclose that she has grant funding for work outside of this project from the Merck Foundation, Genentech and Pfizer. 

All other authors report there are no conflicts of interest."

We note that you received funding from a commercial source: Merck Foundation, Genentech and Pfizer. 

Within this Competing Interests Statement, please confirm that this does not alter your adherence to all PLOS ONE policies on sharing data and materials by including the following statement: ""This does not alter our adherence to PLOS ONE policies on sharing data and materials.” (as detailed online in our guide for authors http://journals.plos.org/plosone/s/competing-interests).  If there are restrictions on sharing of data and/or materials, please state these. Please note that we cannot proceed with consideration of your article until this information has been declared. 

Reviewers' comments:

Reviewer's Responses to Questions

**Comments to the Author**

1. Is the manuscript technically sound, and do the data support the conclusions?

Reviewer #1: Partly

Reviewer #2: Yes

2. Has the statistical analysis been performed appropriately and rigorously? 

Reviewer #1: N/A

Reviewer #2: N/A

3. Have the authors made all data underlying the findings in their manuscript fully available?

Reviewer #1: No

Reviewer #2: Yes

4. Is the manuscript presented in an intelligible fashion and written in standard English?

Reviewer #1: Yes

Reviewer #2: Yes

5. Review Comments to the Author

Reviewer #1: This is an interesting feasibility study looking at provision of weight management support in a rural setting. A key strength of the study is the objective assessment of weight change and physiological outcomes. However, in my opinion there are two weaknesses to the study, which both look to be resolvable: Firstly this is in how the study is described. For reasons articulated in my more detailed comments, I don’t believe this is an acceptability study, or at least the outcomes reported and relied on here do not relate to frameworks of acceptability. So this would need to be acknowledged and adapted. Secondly, the behavioural support is not described to the level currently required - a clearer articulation of the specific behaviour change techniques that have been implemented is needed, mapping these to the intended mediators of behaviour change (e.g., self-efficacy). It looks like a large number of these have been provided - including some in the control group - but unless they are more systematically set out it is hard to work out what the logic model/mechanism of action is for the study, which can then be measured against when it comes to trial.

Abstract:

- I would reframe the 1st sentence - this is contingent on interventions working in the long term which has yet to be established. It’s more accurate to say that weight loss itself does confer the benefits refer to.

- The Method and Results sections were somewhat confusing; the methods talk about 40 people, but the results refer to 423 - what was the sample pool? The Methods should contain a clear account of how you intend to measure feasibility and acceptability (how, and what criteria would constitute a decision either way), and then in the Results report whether these were met.

Method

- Study design; please provide the rationale and justification of the criteria set for feasibility and acceptability. The measure of acceptability in particular is unusual so needs some supporting - for example see the Theoretical framework of Acceptability (Sekhon et al., 2017). This sets out a systematically developed multi-dimensional framework, but it does not include the measure you had (i.e., attendance). To me it therefore comes across that both of the outcomes you report are feasibility, and you haven’t reported a measure of acceptability in this paper.

- P7, sentence that starts “The weight loss was achieved……” - take care not to infer things that you could not control. This was the intention, you can’t claim that this was actually how the weight loss was achieved until you have tested this (i.e., ppts could just reduce calories and not increase PA and still achieve weight loss, etc).

- P7: you refer to providing behavioural techniques. These need to be fully described, I am guessing both in relation to what the telephone counsellor did, and what was provided through the manual. Some elements are referred to later - but it’s hard to follow when definitions, clarifications or signposting to where this is covered in more detail are not provided when first introduced. The full detail then appears in P10. While there is lots of detail here, and it looks like you have linked strategies and techniques to your intended outcomes, it is not presented in a way that is easy to reference for the reader. I would consider finding a way to list/map the intended mediators you want to influence (e.g., self-efficacy) to evidence-based or theoretically informed techniques you have chosen to tackle them. Moving away from description to mapping this more clearly would also help when it comes to assessing mechanisms of action (i.e., whether elements were sufficient to bring about hypothesised changes). Similarly, while you describe familiar approaches to behavioural support, you have not specified the behaviour change techniques used - which is now expected practice for any published intervention. This could be done using the Michie et al Taxonomy of Behaviour Change techniques, which is well developed and implemented in research and practice (a recent review is referred to below for reference).

- P8 - motivational interviewing is introduced as a technique, but the most recent evidence in relation to its efficacy for weight loss is not referenced.

- P9 - you state that “exercise prescriptions were appropriately modified….” (P9).What did this mean? This issue (i.e., stating something before defining it) was evidence throughout and made it quite hard to follow - for example, this was also the case for describing how

- P11 - given recent discussion about the importance of specifying what is in a control group, you could also use the behaviour change taxonomy to indicate what is in the control group (i.e., there seems to be quite a lot they are also receiving). This may help to make it clearer what differs between the two groups.

-

Results

- It would be useful to make some comment on the reasons for ineligibility (which appear in the figure, but aren’t referred to in the text). For e.g., it would be useful to know this for others aiming to recruit this way, in terms of whether the reasons would have been known to potential participants (e.g., different advertising might make a difference) or if the reasons were unknown to those applying.

- I was surprised to see data reported for participants who had not consented to be part of the study (i.e., Table 1).

- The text in the main paragraph of P16 (describing Table 2) is largely repeating the table - only a brief sentence of two is needed to flag the key points to the reader.

Discussion

- P20 - when you refer to strengths, the first paragraph of this did not seem to be supported by any external evidence or what is reported for your study. A lot of claims are made in the first sentence alone - for example you state that you established social support - but this is not something you have assessed. The reference to participant feedback would be fine if this had been systematically collected and reported, but this comes across as a single anecdote.

- Paragraph 2 on P20 is similarly based on feedback from participants that you have not reported - so again, does not provide a convincing case. If you have this qualitative data, then it should be presented as part of the Results before you can refer to it.

- P21 you refer to a pilot study, but this is a feasibility study.

Refs:

Sekhon, M., Cartwright, M., & Francis, J. J. (2017). Acceptability of healthcare interventions: an overview of reviews and development of a theoretical framework. BMC health services research, 17(1), 1-13.

Carey, R. N., Connell, L. E., Johnston, M., Rothman, A. J., De Bruin, M., Kelly, M. P., & Michie, S. (2019). Behavior change techniques and their mechanisms of action: a synthesis of links described in published intervention literature. Annals of Behavioral Medicine, 53(8), 693-707.

Reviewer #2: Abstract

- Results describes the sampling and drop out too extensively. In my view this can just be described in the body of the paper

Introduction

- Line 3 – ‘lower access and fewer resources to healthcare’ clarity needs improving

- The authors make a good case for the project, with strong rationale given for need for WL programme specific to rural settings and the need to increase self-efficacy and social support in rural populations. However, there was limited review of the current literature and other similar studies. The paper would benefit from this as it would provide context for the current study

Methods

Methods were thorough and would enable replication. However, there are some areas that require more clarity:

- Were dietary consultations from the registered dietitian provided in addition to the weekly motivational interviewing approach?

- It is unclear to me whether the behavioural, dietary and physical activity components of the intervention were separately delivered i.e. in separate phone calls, or whether the health coach delivered all elements within one 30-45 minute call.

- Similarly, the group-mediated activity following the weekly telephone counselling that was integrated into the behavioural component – does this not apply to the physical activity and dietary element too?

- Greater clarity would be appreciated here. Perhaps within the first paragraph of the ‘Telephone-based Health Counselling’ section it might be useful to explain the overall intervention in more detail and then use the individual sections to go into the specifics of the dietary/PA requirements.

- The feasibility was determined by participants completing online surveys. Given that this is a rural population, who may have limited access to the internet, were other methods for completing the surveys provided (i.e. pen and paper, or over the telephone)?

Results

- The data presented in Table 2 is presented in the text. You can just refer to the table without repeating the results. It would also be useful to know whether there were any significant differences between groups on these baseline characteristics

- I appreciate the inclusion of reasons for drop out – this provides important context

- Per Table 1 there was a significant difference in BMI between participants who were excluded and consented – some comment or reflection on this in the discussion would be welcomed.

Discussion

- Previous research was mentioned in the discussion, but this had not been previously covered in the introduction. Agree that it is important to contextualise the findings but the paper would benefit from a more thorough literature review in the introduction, which can then be referred to in the discussion.

- Reflection on the impact of COVID was useful and provided additional context to the results

Other comments

- This is a clear and thorough paper. I look forward to reading the results of the intervention.

6. PLOS authors have the option to publish the peer review history of their article (what does this mean?). If published, this will include your full peer review and any attached files.

Reviewer #1: No

Reviewer #2: **Yes: **Jordan Rea Marwood

---

## [Author Response · Author response to Decision Letter 0]

18 Jan 2023

Review Comments to the Author

Reviewer #1: This is an interesting feasibility study looking at provision of weight management support in a rural setting. A key strength of the study is the objective assessment of weight change and physiological outcomes. However, in my opinion there are two weaknesses to the study, which both look to be resolvable: Firstly this is in how the study is described. For reasons articulated in my more detailed comments, I don’t believe this is an acceptability study, or at least the outcomes reported and relied on here do not relate to frameworks of acceptability. So this would need to be acknowledged and adapted. Secondly, the behavioural support is not described to the level currently required - a clearer articulation of the specific behaviour change techniques that have been implemented is needed, mapping these to the intended mediators of behaviour change (e.g., self-efficacy). It looks like a large number of these have been provided - including some in the control group - but unless they are more systematically set out it is hard to work out what the logic model/mechanism of action is for the study, which can then be measured against when it comes to trial.

RESPONSE: Thank you for your comments. We have revised the manuscript to address your concerns regarding the acceptability and behavioral support content. Please see the individual responses below.

Abstract:

- I would reframe the 1st sentence - this is contingent on interventions working in the long term which has yet to be established. It’s more accurate to say that weight loss itself does confer the benefits refer to.

RESPONSE: We revised the abstract as requested. We specified that weight loss through lifestyle modification can produce health benefits to align with the scope of this study.

- The Method and Results sections were somewhat confusing; the methods talk about 40 people, but the results refer to 423 - what was the sample pool? The Methods should contain a clear account of how you intend to measure feasibility and acceptability (how, and what criteria would constitute a decision either way), and then in the Results report whether these were met.

RESPONSE: We clarified the methods and results sections as requested. We had 423 individuals enter the online screening survey providing consent upon doing so. Of these, 98 were determined to be eligible based on screening procedures. Ultimately, 40 subjects were enrolled in the study and randomized to one of the two conditions. 

Thank you for the suggestions regarding the framework of acceptability. Given what was measured and reported in this study, we have reviewed all uses of the term “acceptability” and changed text to more appropriate term of “adherence”. 

In the methods section, we defined the feasibility “at least 80% participants completing the follow-up surveys”. The acceptable adherence target to the weight loss intervention was defined as, “Percentage of participants in the weight loss group who attend ≥75% of weekly telephone sessions.” The results section states, “Feasibility was demonstrated,” which refers to the feasibility target being met. We also reported that “the adherence of the intervention was acceptable among half of the group”.

Methods

- Study design; please provide the rationale and justification of the criteria set for feasibility and acceptability. The measure of acceptability in particular is unusual so needs some supporting - for example see the Theoretical framework of Acceptability (Sekhon et al., 2017). This sets out a systematically developed multi-dimensional framework, but it does not include the measure you had (i.e., attendance). To me it therefore comes across that both of the outcomes you report are feasibility, and you haven’t reported a measure of acceptability in this paper.

RESPONSE: Thank you for the suggestions. Please refer to our previous response regarding changes made to better reflect appropriate terminology and domains of intervention adherence and feasibility. We revised the methods section to reflect the changes (page 7, page 18)

- P7, sentence that starts “The weight loss was achieved……” - take care not to infer things that you could not control. This was the intention, you can’t claim that this was actually how the weight loss was achieved until you have tested this (i.e., ppts could just reduce calories and not increase PA and still achieve weight loss, etc).

RESPONSE: We revised the sentence as, “The targeted behavioral modifications for weight loss included…”(page 9, last paragraph)

- P7: you refer to providing behavioural techniques. These need to be fully described, I am guessing both in relation to what the telephone counsellor did, and what was provided through the manual. Some elements are referred to later - but it’s hard to follow when definitions, clarifications or signposting to where this is covered in more detail are not provided when first introduced. The full detail then appears in P10. While there is lots of detail here, and it looks like you have linked strategies and techniques to your intended outcomes, it is not presented in a way that is easy to reference for the reader. I would consider finding a way to list/map the intended mediators you want to influence (e.g., self-efficacy) to evidence-based or theoretically informed techniques you have chosen to tackle them. Moving away from description to mapping this more clearly would also help when it comes to assessing mechanisms of action (i.e., whether elements were sufficient to bring about hypothesised changes). Similarly, while you describe familiar approaches to behavioural support, you have not specified the behaviour change techniques used - which is now expected practice for any published intervention. This could be done using the Michie et al Taxonomy of Behaviour Change techniques, which is well developed and implemented in research and practice (a recent review is referred to below for reference).

RESPONSE: We appreciate your suggestions regarding more contemporary and precise reporting of the active elements of intervention and hypothesized mechanisms of action. We have now included a BCT-MoA (Behavior Change Techniques-Mechanisms of Action) table including intervention applications, behavior change techniques used, and links to hypothesized mechanisms of action for the telephone-based health counseling group (Supplementary Table 2). In addition to the BCT-MoA table, we revised the intervention approach substantially (P8-10, P11-14). 

- P8 - motivational interviewing is introduced as a technique, but the most recent evidence in relation to its efficacy for weight loss is not referenced.

RESPONSE: This is a good point. We’ve added references demonstrating the efficacy of motivational interviewing in weight management studies (see page 11, first paragraph), and also cited a recent large-scale weight loss intervention trial that applied motivational interviewing (Page 11, Reference 34). 

- P9 - you state that “exercise prescriptions were appropriately modified….” (P9). What did this mean? This issue (i.e., stating something before defining it) was evidence throughout and made it quite hard to follow - for example, this was also the case for describing how

RESPONSE: Thank you for the comment. We understand the importance of specifying “modifications” and “individualized approach” throughout the manuscript. We added examples of how and why the exercise prescription was modified (page 11-12). Since the weight loss intervention was a translational, telephone-based, one-on-one intervention, and health coaches individualized physical activity, diet, and behavioral components based on the specific needs for each participant, while gradually progressing towards the specified global volume goals, we are not able to provide a more general exercise prescription than has been described. We content this personalization process represents a strength of the intervention as it is well-established “one-size fits all” prescriptions are not efficacious. For the behavioral support component, along with utilizing individualized counseling, we did specify that the weekly telephone session was primarily focused on the planned lessons and followed a standard session agenda from the lifestyle modification manual. Thus, the intervention for the weight loss group was structured, evidence-based, and included the implementation of strategies used to tailor content/interactions to participant needs; a particularly unique aspect of this study.

- P11 - given recent discussion about the importance of specifying what is in a control group, you could also use the behaviour change taxonomy to indicate what is in the control group (i.e., there seems to be quite a lot they are also receiving). This may help to make it clearer what differs between the two groups.

RESPONSE: Thank you for the comment. To clarify the difference between the two groups, we added the table below as a supplementary table (Supplementary Table 1). However, we did not map BCTs for the control group since those were resources provided without additional advice, instruction, or demonstration from health coaches (Michie et. al., 2013). 

Supplementary Table 1. Study materials and supplementary resources by study group

 Telephone-based Health Counseling Health Education

Materials provided 

at baseline • A lifestyle modification manual 

• Exercise manual and online videos 

• Fitbit tracker 

• Weight scale 

• Self-log • Education brochure (AICR guideline for physical activity and diet) 

• Exercise manual and online videos 

• Self-log 

Other components provided 15 weekly telephone sessions 

• Health coach - provided education & counseling sessions 

• Weekly lifestyle education lessons 

• Tailored dietary modification, aerobic & resistance exercise recommendations 

• Individualized behavioral counseling - self-regulation strategies, social support & managing barriers 

• Group messages At the end of the study 

• A lifestyle modification manual 

• Fitbit tracker 

Results

- It would be useful to make some comment on the reasons for ineligibility (which appear in the figure, but aren’t referred to in the text). For e.g., it would be useful to know this for others aiming to recruit this way, in terms of whether the reasons would have been known to potential participants (e.g., different advertising might make a difference) or if the reasons were unknown to those applying.

RESPONSE: Thank you for the comment. We opted not to include the details of ineligibility in the text since that was not the focus of this manuscript. As you mentioned, we have detailed information in the figure for readers who were interested in the recruitment. As mentioned in the text, most individuals were recruited through Facebook (n= (384/423)). Due to the small proportion of individuals recruited by other methods, we were not able to compare the differences in the ineligibility across recruitment approach. 

- I was surprised to see data reported for participants who had not consented to be part of the study (i.e., Table 1).

RESPONSE: We revised the process of recruitment in the methods section, “Individuals who were interested in the study were consented to complete an eligibility screening,” to clarify that all individuals who entered the survey consented to the online survey. We revised the title of Table 1 to “Characteristics of Individuals Who Completed the Online Screening Survey, Overall and Those Who Were Consented to the Study vs. Those Were Screened But Excluded from the Study”. Table 1 is critical to include to show the differences between those who consented and were subsequently enrolled and randomized, versus those who were excluded from the weight loss study. We revised the results section (P17) to reflect the changes. This is an important aspect to illustrate the generalizability of the study sample, and also crucial for comparison with other studies. 

- The text in the main paragraph of P16 (describing Table 2) is largely repeating the table - only a brief sentence or two is needed to flag the key points to the reader.

RESPONSE: We revised as requested.

Discussion

- P20 - when you refer to strengths, the first paragraph of this did not seem to be supported by any external evidence or what is reported for your study. A lot of claims are made in the first sentence alone - for example you state that you established social support - but this is not something you have assessed. The reference to participant feedback would be fine if this had been systematically collected and reported, but this comes across as a single anecdote.

RESPONSE: In this paragraph, we intended to highlight what was unique about our study, which was tailoring the intervention to a rural population. We have edited certain language (i.e., “establishing”) in this section to reflect the change to “promoting” as part of the intervention. Given the focus of this study was on the feasibility of providing individualized counseling to participants and collecting data in the community setting, we believe the slight modification to this language now accurately reflects procedures while avoiding making any unjustified claims. 

We did provide behavioral support with the intention of influencing theoretical determinants of behavior change (e.g., social support). However, formal analysis of potential mediators of behavioral changes was beyond the scope of the current study. We acknowledge that participant feedback was not systematically assessed. Thus, we did not include their feedback in the results section. We mentioned the anecdotal feedback in the discussion section to provide an example of how counseling was utilized in the intervention to tailor recommendations to address individual barriers. 

- Paragraph 2 on P20 is similarly based on feedback from participants that you have not reported - so again, does not provide a convincing case. If you have this qualitative data, then it should be presented as part of the Results before you can refer to it.

RESPONSE: We agree that this was not a qualitative study, therefore, we did not include participant feedback in the results section. These anecdotal examples from participants outline the importance of our tailored approach to address the unique needs of a rural population and provide meaningful context.

- P21 you refer to a pilot study, but this is a feasibility study.

RESPONSE: This was a feasibility study, and based-on the sample size, this was also a pilot study. We did not intend to interchange feasibility with pilot. But in this paragraph, we used “pilot study” to refer our small sample size and indicate that a large-scale study is needed. A pilot study is defined as “A small-scale test of the methods and procedures to be used on a larger scale” (Porta, Dictionary of Epidemiology, 5th edition, 2008). Pilot studies should assess the feasibility/acceptability of the approach to be used in the larger study (https://www.nccih.nih.gov/grants/pilot-studies-common-uses-and-misuses).

Refs:

Sekhon, M., Cartwright, M., & Francis, J. J. (2017). Acceptability of healthcare interventions: an overview of reviews and development of a theoretical framework. BMC health services research, 17(1), 1-13.

Carey, R. N., Connell, L. E., Johnston, M., Rothman, A. J., De Bruin, M., Kelly, M. P., & Michie, S. (2019). Behavior change techniques and their mechanisms of action: a synthesis of links described in published intervention literature. Annals of Behavioral Medicine, 53(8), 693-707.

Reviewer #2: Abstract

- Results describes the sampling and drop out too extensively. In my view this can just be described in the body of the paper

RESPONSE: We revised as requested on page 3, results paragraph.

Introduction

- Line 3 – ‘lower access and fewer resources to healthcare’ clarity needs improving

RESPONSE: We revised the introduction as requested. We restructured the first paragraph to point out the obesity-related disparities in rural residents, and then clarified the socioeconomic and environmental disadvantages that are linked with health disparities in this population (see page 5, first paragraph). 

- The authors make a good case for the project, with strong rationale given for need for WL program specific to rural settings and the need to increase self-efficacy and social support in rural populations. However, there was limited review of the current literature and other similar studies. The paper would benefit from this as it would provide context for the current study.

RESPONSE: Thank you for the comment and suggestion for a more substantive review of contemporary literature. The Introduction section has been revised to provide more review of the salient research beginning on page 6, first paragraph.

Methods

Methods were thorough and would enable replication. However, there are some areas that require more clarity:

- Were dietary consultations from the registered dietitian provided in addition to the weekly motivational interviewing approach?

RESPONSE: Yes. We clarified this in the methods and discussion sections (see page 11, second paragraph and page 26, second paragraph), referring to consultations being provided upon request on an as needed basis.

- It is unclear to me whether the behavioural, dietary and physical activity components of the intervention were separately delivered i.e. in separate phone calls, or whether the health coach delivered all elements within one 30-45 minute call.

RESPONSE: The behavioral, dietary and physical activity counseling addressed each of these domains during the weekly telephone sessions. During these sessions, instructions on and recommendations for independent dietary and physical activity were made along with instruction on complementary self-regulation strategies. We revised the methods section to clarify this information (Page 8-9). To improve clarity, we have included a supplementary table defining all applications for the health counseling group, behavior change techniques (according to the BCTTv1), and linking BCTs to hypothesized mechanisms of action. We revised the methods section to reflect this change (Page 12).

- Similarly, the group-mediated activity following the weekly telephone counselling that was integrated into the behavioural component – does this not apply to the physical activity and dietary element too?

RESPONSE: The group chat was provided as an ancillary supportive piece of the telephone-based health counseling intervention and available for participants in that arm of the study. The chat was supervised by the health coaches and primarily used as a means of communication between participants. Additionally, it was used to facilitate group-mediated barrier problem solving and as a means for participants to share successes and challenges and provide encouraging feedback to their peers. We revised the methods section to reflect this change (Page 13, second paragraph).

- Greater clarity would be appreciated here. Perhaps within the first paragraph of the ‘Telephone-based Health Counselling’ section it might be useful to explain the overall intervention in more detail and then use the individual sections to go into the specifics of the dietary/PA requirements.

RESPONSE: Thank you for the suggestion to elaborate on the intervention components. To improve clarity, we have included additional details in the first paragraph of the Telephone-based health counseling section, as well as included a supplementary table describing the intervention applications by component with the relevant behavior change techniques and hypothesized mechanisms of action (Page 8-9). We hope this clarifies now the dietary, physical activity, and behavioral support components of the counseling sessions were addressed.

- The feasibility was determined by participants completing online surveys. Given that this is a rural population, who may have limited access to the internet, were other methods for completing the surveys provided (i.e. pen and paper, or over the telephone)?

RESPONSE: We understand your concern regarding accessibility of internet in rural populations. Because most participants were recruited through social media, and all individuals went through online screening surveys before enrolling in the study, participants who enrolled in the study would have internet access to the online survey. We did offer the eligibility screening through telephone, but no one requested a telephone screening due to lack of access to internet. Thus, we did not offer any follow up surveys through paper copies or telephone. 

Results

- The data presented in Table 2 is presented in the text. You can just refer to the table without repeating the results. It would also be useful to know whether there were any significant differences between groups on these baseline characteristics

RESPONSE: We shortened the text describing results in Table 2. We decided to not compare baseline characteristics between group. Since this was a randomized controlled study, the study design ensures that the two groups are balanced in respect of baseline characteristics and unmeasured confounders. 

- I appreciate the inclusion of reasons for drop out – this provides important context

RESPONSE: Thank you.

- Per Table 1 there was a significant difference in BMI between participants who were excluded and consented – some comment or reflection on this in the discussion would be welcomed.

RESPONSE: Thank you for the comment. Although the magnitude of the BMI differed between the 2 groups, the difference did not reach the statistical significance. However, we revised the discussion section to reflect individuals who enrolled in the study may not represent the general population in rural areas (page 24, last paragraph).

Discussion

- Previous research was mentioned in the discussion, but this had not been previously covered in the introduction. Agree that it is important to contextualise the findings but the paper would benefit from a more thorough literature review in the introduction, which can then be referred to in the discussion.

RESPONSE: Thank you for the comment and suggestion for a more substantive review of contemporary literature. The Discussion section has been revised.

- Reflection on the impact of COVID was useful and provided additional context to the results

RESPONSE: Thank you. We wanted to acknowledge the challenges of COVID-19 and share our experience with all readers.

Other comments

- This is a clear and thorough paper. I look forward to reading the results of the intervention.

RESPONSE: We appreciate your inputs for this paper. We are currently in the process of drafting another manuscript to show the changes in weight, body composition, and disease biomarkers.

---

## [Decision Letter · Decision Letter 1]

22 Feb 2023

The Feasibility of a Telephone-based Weight Loss Intervention in Rural Ohio: A Pilot Study

PONE-D-22-26251R1

Dear Dr. Zhang,

We’re pleased to inform you that your manuscript has been judged scientifically suitable for publication and will be formally accepted for publication once it meets all outstanding technical requirements.

Kind regards,

Jamie Matu, Ph.D.

Academic Editor

PLOS ONE

**Comments to the Author**

Reviewer #2: All comments have been addressed

---

## [Editor Report · Acceptance letter]

6 Mar 2023

PONE-D-22-26251R1 

The Feasibility of a Telephone-based Weight Loss Intervention in Rural Ohio: A Pilot Study 

Dear Dr. Zhang:

I'm pleased to inform you that your manuscript has been deemed suitable for publication in PLOS ONE. Congratulations! Your manuscript is now with our production department. 

Kind regards, 

on behalf of

Dr. Jamie Matu 

Academic Editor

PLOS ONE